# Differential effects of follicle-stimulating hormone glycoforms on the transcriptome profile of cultured rat granulosa cells as disclosed by RNA-seq

Teresa Zariñán[1], Jesús Espinal-Enriquez[2], Guillermo De Anda-Jáuregui[2], Saúl Lira-Albarrán[3], Georgina Hernández-Montes[1], Rubén Gutiérrez-Sagal[1], Rosa G. Rebollar-Vega[1], George R. Bousfield[4], Viktor Y. Butnev[4†], Enrique Hernández-Lemus[2], Alfredo Ulloa-Aguirre[1]*

**1** Red de Apoyo a la Investigación, Universidad Nacional Autónoma de México (UNAM)-Instituto Nacional de Ciencias Médicas y Nutrición SZ, Mexico City, Mexico, **2** Instituto Nacional de Medicina Genómica (INMEGEN), Mexico City, Mexico, **3** Department of Reproductive Biology, Instituto Nacional de Ciencias Médicas y Nutrición Salvador Zubirán, Mexico City, Mexico, **4** Department of Biological Sciences, Wichita State University, Wichita Kansas, Kansas, United States of America

☯ These authors contributed equally to this work.
† Deceased.
¤ Current address: Departamento de Gestión Académica e Investigación, Hospital Escuela, Tegucigalpa, Francisco Morazán, Honduras
* alfredo.ulloaa@incmnsz.mx, aulloaa@unam.mx

## Abstract

It has been documented that variations in glycosylation on glycoprotein hormones, confer distinctly different biological features to the corresponding glycoforms when multiple *in vitro* biochemical readings are analyzed. We here applied next generation RNA sequencing to explore changes in the transcriptome of rat granulosa cells exposed for 0, 6, and 12 h to 100 ng/ml of four highly purified follicle-stimulating hormone (FSH) glycoforms, each exhibiting different glycosylation patterns: *a*. human pituitary FSH[18/21] (hypo-glycosylated); *b*. human pituitary FSH[24] (fully glycosylated); *c*. Equine FSH (*eq*FSH) (hypo-glycosylated); and *d*. Chinese-hamster ovary cell-derived human recombinant FSH (*rec*FSH) (fully-glycosylated). Total RNA from triplicate incubations was prepared from FSH glycoform-exposed cultured granulosa cells obtained from DES-pretreated immature female rats, and RNA libraries were sequenced in a HighSeq 2500 sequencer (2 x 125 bp paired-end format, 10–15 x 10⁶ reads/sample). The computational workflow focused on investigating differences among the four FSH glycoforms at three levels: gene expression, enriched biological processes, and perturbed pathways. Among the top 200 differentially expressed genes, only 4 (0.6%) were shared by all 4 glycoforms at 6 h, whereas 118 genes (40%) were shared at 12 h. Follicle-stimulating hormone glycocoforms stimulated different patterns of exclusive and associated up regulated biological processes in a glycoform and time-dependent fashion with more shared biological processes after 12 h of exposure and fewer treatment-specific ones, except for *rec*FSH, which exhibited stronger responses with more specifically associated processes at this time. Similar results were found for down-regulated processes, with a

**Data Availability Statement:** All relevant data are within the paper and its Supporting information files. All statistical database are available at https://osf.io/57jf/.

**Funding:** This study was supported by grants from CONACyT, Mexico (grant no. 240619) and the Coordinación de la Investigación Científica-UNAM, Mexico (to A.U-A). G.R.B. and VYB were supported by NIH grant P01AG-029531". The funders had no role in study design, data collection and analysis, decision to publish, or preparation of the manuscript.

**Competing interests:** The authors have declared that no competing interests exist.

greater number of processes at 6 h or 12 h, depending on the particular glycoform. In general, there were fewer downregulated than upregulated processes at both 6 h and 12 h, with FSH[18/21] exhibiting the largest number of down-regulated associated processes at 6 h while *eq*FSH exhibited the greatest number at 12 h. Signaling cascades, largely linked to cAMP-PKA, MAPK, and PI3/AKT pathways were detected as differentially activated by the glycoforms, with each glycoform exhibiting its own molecular signature. These data extend previous observations demonstrating glycosylation-dependent distinctly different regulation of gene expression and intracellular signaling pathways triggered by FSH in granulosa cells. The results also suggest the importance of individual FSH glycoform glycosylation for the conformation of the ligand-receptor complex and induced signalling pathways.

## Introduction

Follicle-stimulating hormone is produced by the anterior pituitary gland (AP) as different glycoforms, defined by the presence or abscence of glycans on the hormone-specific β-subunit. As other glycoprotein hormones, this gonadotropin is composed of two subunits, the α and β subunits, associated with each other through non-covalent interactions; the α-subunit, is common to all glycoprotein hormones [luteinizing hormone (LH), choriogonadotropin (CG) and thyroid-stimulating hormone (TSH)], whereas the β-subunit, confers specificity for binding and action to each gonadotropin at its cognate receptor [1]. In human FSH (hFSH), four Asn residues, two in FSHα (αN52 and αN78) and two in FSHβ (βN7 and βN24) are targets for N-linked glycosylation [2] (Fig 1). In glycoprotein hormones, the oligosaccharide in position $\alpha Asn^{52}$ is involved in the activation of the receptor/signal transducer (G protein) system and biological response [3–6], an effect mediated through the stabilization of the conformation of the FSH dimer bound to the FSH receptor (FSHR) [7–9]. Meanwhile, glycans attached to FSHβ play a major role in defining the circulatory half-life and *in vivo* bioactivity of the gonadotropin [10], albeit it has been shown that they also impact on FSH-mediated intracellular signaling [11–14].

Human FSH macroheterogeneity occurs at either one or both FSHβ N-glycosylation sites (Fig 1), determining differences in apparent molecular weights by gel electrophoresis and immunoblotting [15–17]. In fact, Western blots of FSH recovered after gel electrophoresis, yielded two FSHβ bands, a 24 kDa band exhibiting both FSHβ-subunit $Asn^7$ and $Asn^{24}$ N-linked glycans (which corresponds to fully- o tetra-glycosylated FSH heterodimer or $FSH^{24}$) and a 21kDa band in which the $Asn^{24}$ glycan is absent (hypo- or tri-glycosylated FSH heterodimer or $FSH^{21}$) [15, 18]. Purified Hypo-glycosylated preparations include an additional hypo-glycosylated variant lacking the $Asn^7$ glycan in FSHβ, with its corresponding heterodimer designated as $FSH^{18}$ (18 kDa-FSH) (Fig 1c), hence the designation FSH[18/21]. Although an additional hypo-glycosylated FSH variant also has been detected in human pituitaries (exhibiting a 15 kDa-FSHβ), the corresponding β-subunit assembles poorly with FSHα and consequently very low levels of secretion of the corresponding heterodimer ($FSH^{15}$ glycoform) are detected [19]. Thus, three hFSH glycoforms ($FSH^{24}$, $FSH^{21}$, and $FSH^{18}$) appear to be the physiologically relevant FSH variants in humans.

Similar to the influence of microheterogeneity (*i.e.* variations in the structure of the carbohydrates and complexity of the oligosaccharides on gonadotropins) on the ability of the gonadotropins to activate the FSHR and trigger intracellular signaling [22–24], FSH macroheterogeneity also contributes to its bioactivity [14, 25, 26]. In fact, *in vitro* and *in vivo*

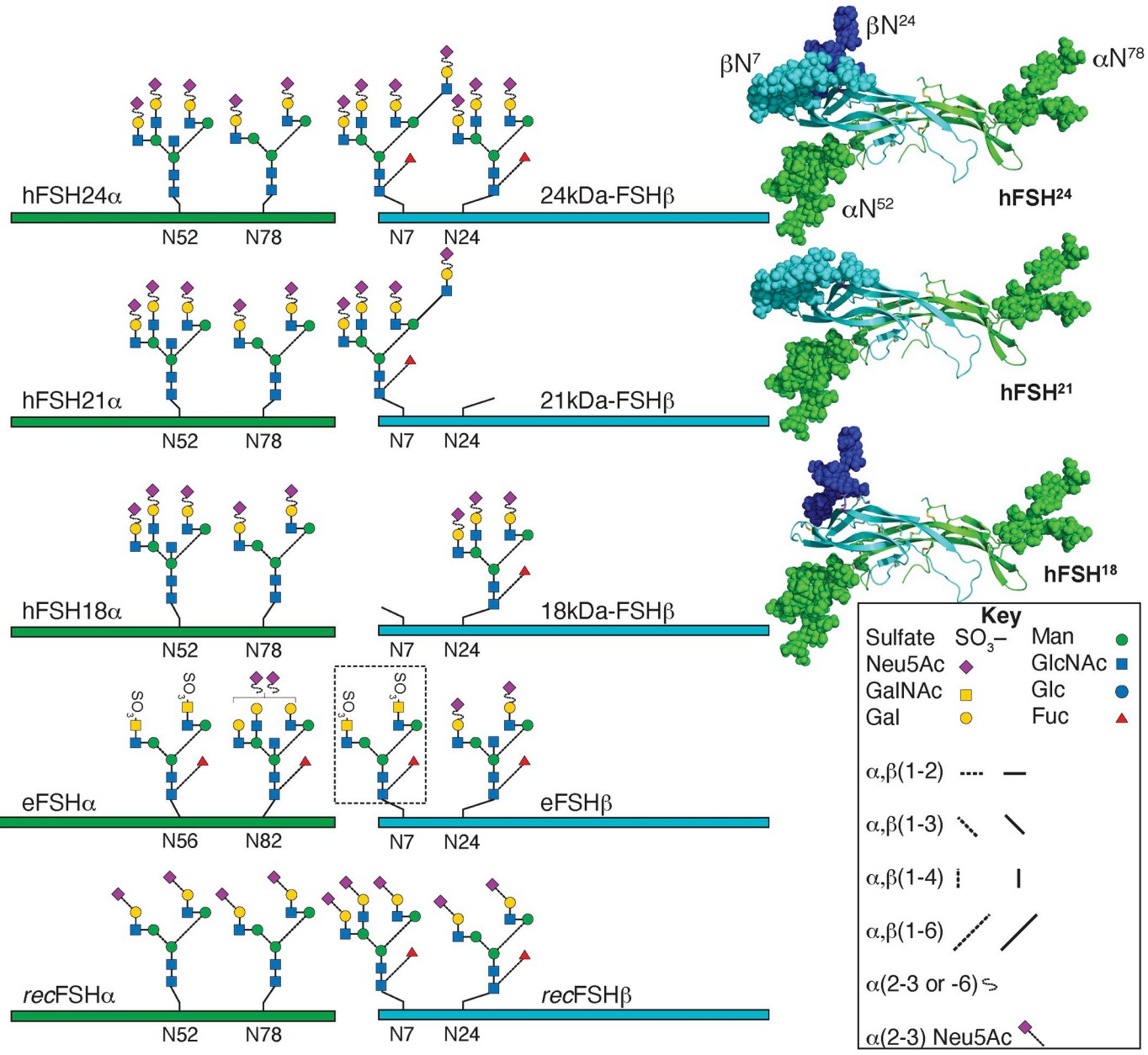

**Fig 1. Typical glycans attached to human pituitary FSH (hFSH) glycoforms, human recombinant FSH produced by Chinese hamster ovary cells (recFSH), and equine FSH (eFSH).** The green and cyan bars and ribbons (3-D structures at right) indicate the common-α and hormone-specific FSHβ subunits, respectively. N-glycosylation sites are indicated by the numbers below the bars. eFSHα subunit has 4 additional amino acid residues at the N-terminus, accounting for the difference in numbering. The glycan at position β7 in eqFSH (black dotted square) is absent in the bulk (90%) of the molecules contained in highly purified preparations [20]. Glycans in recFSH were taken from Mastrangeli et al [21]. Human FSH glycoform models created with the GLYCAM web tool are shown on the right with the subunits rendered as cartoons using PyMol. The same color scheme is employed for subunits and glycans, which are rendered as spheres. Partially obscured Asn[24] glycan is colored dark blue to distinquish it from Asn[7] glycan (light blue).

sgtudies have shown that pituitary and recombinant hFSH glycoforms display differential FSHR binding kinetics and bioactivity [12, 14, 27, 28]. Further, in a recent study employing cAMP accumulation, ?-arrestin-mediated ERK1/2 activation, and intracellular calcium (iCa2+) accumulation as read-outs in FSH-stimulated HEK-293 cells, we found that in addition to

determining the intensity of the biological response at the target cell, the presence or absecnce of glycans attached to FSHβ conferred some degree of biased agonism to the different FSH glycoforms [26].

Follicle-stimulating hormone stimulation triggers activation of a complex array of diverse signaling cascades mediated not only by the canonical Gs/cAMP/PKA pathway but also by other G proteins and receptor interacting proteins [29, 30]. Activation of this signaling network and particular signaling modules most probably occurs through stabilization of different FSHR conformations by FSH glycoforms possessing distinct glycosylation patters [28]. Given the differential effects of FSH glycoforms on particular FSH-stimulated read outs [26], we here applied next generation sequencing (NGS) to assess as primary objective whether different FSH perturbogens exhibiting differential glycosylation led to distinct gene expression patterns and biological processes across time points, which might allow to better understand the physiological relevance of differential FSH glycosylation (particularly on those naturally occurring variants synthesized and secreted by the anterior pituitary gland) during the human menstrual cycle. To this end, cultured rat granulosa cells were exposed during different times to a fixed dose of four highly purified FSH glycoform preparations, each exhibiting different glycosylation patterns: *a*. human pituitary FSH[18/21] (hypo-glycosylated); *b*. human pituitary FSH[24] (tetra- or fully-glycosylated); *c*. Equine FSH (*eq*FSH, 90% hypo-glycosylated); and *d*. Chinese-hamster ovary cell-derived human recombinant FSH (*rec*FSH; 80% tetra-glycosylated) (Fig 1). These four preparations also differ to varying extent in microheterogeneity [15, 20, 25, 26, 31–38]. Our analysis was focused on the effects of each FSH preparation at three levels: gene expression, enriched biological processes, and perturbed pathways.

## Material and methods

### Hormones

Human pituitary FSH[24] and FSH[18/21] were purified after extraction of human pituitary tissue (a generous gift of Dr. James A. Dias, University at Albany, Albany, NY, USA), employing Superdex G75 chromatography of fractions obtained after Sephacryl S-100 and immunoaffinity chromatography, as described in detail previously [24]. The fractions possessing largely 24 kDa-FSHβ were pooled to generate FSH[24] and those possessing largely 18kDa- and 21 kDa-FSHβ were pooled to obtain FSH[18/21]. Recombinant human FSH produced in Chinese hamster ovary cells (Follitropin Alfa, batches AU012310 and BA024393), was a gift of Merck Serono (Mexico City, Mexico); according to the manufacturer [34], the batch-to-batch consistency for this recombinant FSH compound exhibits coefficients of variation that ranges from 7% to 15% for its most and least abundant isoforms. Equine FSH (batch VB-I-171) was purified from horse pituitaries obtained from Animal Technologies, Inc., (Tyler, TX, USA), as previously described [39]. Kds for the FSH preparations employed in the present study have been reported previously [26].

### Rat granulosa cells culture

Granulosa cells (GC) from immature (21 days old) Wistar rats pretreated for 4 days with 10 mg diethylstilbestrol (DES) through an implanted 10 mm x 1.5 mm silastic capsule, were collected and cultured following the method described by Jia and Hsue [40] with some modifications [41]. Briefly, rats were anesthetized with 120 mg/kg pentobarbital administered IP, and rapidly euthanized by cervical dislocation performed by a trained technician, followed by laparotomy to remove their ovaries. Granulosa cells were then collected by puncturing the ovarian follicles, pooled, counted, and added to 6-well (34 mm diameter) polystyrene culture dishes (Nunc, Roskilde, Denmark) at a density of 1.0 x 10^6 viable cells/well in serum- and insulin-free

2 ml McCoy's 5a medium (Life Technologies, Grand Island, NY, USA), pH 7.0, supplemented with 2 nM glutamine (Sigma-Aldrich Inc., St. MO, USA), antibiotic (penicillin *plus* streptomycin) reagent (Invitrogen, Waltham, MA, USA), and 0.1% bovine serum albumin (Sigma), and cultured for 24h at 37ºC in a humidified atmosphere of 95% air-5% $CO_2$. At the end of the culture period, cells were washed, redissolved in insulin- and serum-free, supplemented McCoy's 5a medium, incubated for an additional 24h in the absence of androgens, and then exposed during 0, 6, and 12 h to 100 ng/ml of the different FSH preparations or control medium (no FSH) in triplicate wells for each preparation and incubation time. The 100 ng/ml dose was chosen based on preliminary experiments assessing the aromatization response of cultured GC to increasing doses of each FSH preparation, in which maximal estrogen production was achieved with the 100 ng/ml dose (not shown). The project was approved by the Internal Research Committee for the Care and Use of Laboratory Animals of the Instituto Nacional de Ciencias Médicas y Nutrición Salvador Zubirán (Project CINVA-RAI-1864-16/19-1).

### RNA isolation and sequencing

Total RNA was isolated from individual GC wells using the TRIzol TM reagent (Thermo Fisher Scientific, Waltham, MA USA)) and the Direct-zol RNA kit (Zymo Research, Irving, Ca, USA), following the instructions provided by the manufacturer [42]. Total RNA concentration was assessed using the NanoDrop 2000 spectrophotometer (Thermo Fisher Scientific, Waltham, MA, USA) and the quality of RNA in each sample was determined in an Agilent 2100 Bioanalyzer (Agilent Technologies, Santa Clara, CA, USA). Only samples with a RNA Integrity Number (RIN) >8.0 were used. RNA libraries were prepared from 500 ng RNA samples using the TruSeq Stranded mRNA Kit (Illumina Inc., San Diego, CA, US) following the manufacturer's instructions. Libraries were sequenced using an Illumina HiSeq2500 equipment (Illumina) in paired-end (2x125 pb) read runs. Depth of sequencing was 10–15 million reads.

### Data bioinformatics analysis

The main objective of the computational pipeline was to identify differences among the four FSH glycoforms tested at three distinct levels: *a*. gene expression; *b*. active biological processes; and *c*. perturbed pathways. To achieve this goal, we applied three different computational approaches: differential gene expression, over-representation analysis and pathway perturbation analysis.

**Data pre-processing.** Prior to data analysis, a series of pre-processing procedures were applied to the data in order to eliminate/diminish technical errors, biases and other sources of undesired variation Quality parameters of the FASTQ files were checked using the quality control FASTQC tool (https://www.bioinformatics.babraham.ac.uk/projects/fastqc/). FASTQ reads were filtered and trimmed using the AfterQC tool [43]; default parameters were used to discard low-quality reads, trim adaptor sequences, and to eliminate poor-quality bases. The reads passing quality control parameters were aligned and quantified to the *Rattus norvegicus* reference transcriptome Rnor_6.0 using Salmon [44] version 0.8.2. Transcript-level quantifications were imported using the tximport package [45] version 1.22.0.

**Differential expression.** Differential gene expression analysis was performed with the edgeR package [46], by comparing each FSH treatment at 0, 6, and 12 h. Since the number of replicates was limited, it was decided to conserve the top-200 differentially expressed genes (FDR < 0.01). Additionally, the similarities and differences between groups of genes reported as differentially expressed were compared under conditions: *a*. All FSH glycoform preparations at the same incubation time; and *b*. Different incubation times (0, 6, and 12 h) for the

same FSH glycoform (*eg. eq*FSH, *rec*FSH, FSH[18/21], and FSH[24]). Each of these comparisons followed distinct scientific questions. On the one hand, the first comparison was useful to analyze the behavior of granulosa cells under the action of four distinctly different FSH glycoforms, on the other hand the latter comparison served to analyze the dynamics of a given FSH glycoform at different times, providing a set of snapshots of the temporal effects of the molecules.

**Over-representation analysis.** By taking separately the overexpressed and underexpressed gene sets, over-representation analysis to detect whether the differentially expressed genes for each contrast were associated to a specific set of biological processes was performed. We implemented independent analyses for over- and underexpressed genes, based on the idea that sets of overexpresed genes may exacerbate a given process. Conversely, underexpressed genes may indicate a depleted/diminished behavior in a particular biological event. We used the Gene Ontology and the Kyoto Encyclopedia of Genes and Genomes (KEGG) [47] as databases for biological categories. Significance threshold was set at p<0.01 to consider a biological category as significant.

**Pathway perturbation analysis.** Pathway perturbation was assessed using the gene expression data previously described. A subset of KEGG pathways was selected based on the information previously described [29]. Pathways were collected using the Graphite package [48]. Contrasts were made between the experimental conditions described above and the baseline cell culture using the Generally Applicable Gene-set Enrichment for Pathway Analysis (GAGE) package [49]. Multiple test correction was performed using the Benjamini-Hochberg *post hoc* method [50]. Top 4 pathways by q-value (corrected p value) for each contrast were selected, and their network representation was merged into a single metapathway. Betweenness centrality (a measure for the relative importance of molecules for the communication in the corresponding pathways) for each gene in the metapathway was calculated. For each contrast, key genes were identified as those that had above-median logfold change in the differential expression analysis, and above-median betweenness centrality in the metapathway network.

## Validation of FSH glycoforms-sensitive transcripts by real time (RT)-PCR

To validate differentially expressed genes in the RNAseq data, we analysed the mRNA level by RT-PCR of five selected FSH glycoform-sensitive genes (*Pld1*, *Npy1R*, *Amh*, *Vegf-B*, and *Bcl2l1*). cDNA synthesis was performed in 2.5 μg total RNA with 260/280 nm ratio >1.8 following the manufacturer'as instructions for the Maxima First Strand cDNA Synthesis Kit for RT-qPCR (Thermo Fisher), employing random primers and oligo dT.

RT-PCR was performed employing 10 ng cDNA in a 20 μl reaction mixture containing 1 μl TaqMan Gene Expression Assay[®], 1μl TaqMan Gene Expression Assay for Actin or GAPDH reference gene, and 10 μl Master Mix TaqMan Universal PCR[®] (all from Applied Biosystems, Waltham, MA, USA). All reactions were performed under the following conditions: 2 min at 50°C (UNG incubation), 10 min at 95°C (AmpliTaq Gold activation), followed by 40 cycles at 95°C for 15 sec (denaturation) and at 60°C for 1 min (anealing and amplification), employing a Step One plus thermocycler (Applied Biosystems). The results were normalized and expressed as fold changes in gene expression levels relative to control cDNA ($2^{-\Delta\Delta CT}$ method). The TaqMan gene expression assays employed were: Rn01493709_m1, Rn02769337_s1, Rn00563731_g1, Rn01454585_g1, Rn06267811_g1 (for the above described test genes, respectively) and Rn01426628_g1 and Rn01775763_g1 (for actin and GAPDH, respectively). Differences in expression levels of selected genes expressed in response to FSH stimulation as assesed by RT-PCR, were analyzed employing the Student's *t* test at an alpha level for significance <0.05.

## Results

### General contrast between 6 and 12 hours

By comparing the four glycoforms at 6 and 12 h *vs* control (no FSH added) it was possible to determine how many genes were shared among these glycoforms. Fig 2A and 2B show the top200 differentially expressed genes for the four glycoforms at 6 h (A) and 12 h (B). The whole set of differentially expressed genes for each glycoform and contrasts can be found in the S1 to S4 Tables. This analysis allowed to identify particular changes in gene expression induced by each glycoform. At 6 h, the number of genes shared among the glycoforms (center of Fig 2A) were only 4 genes, whereas at 12 h this number increased to 118 genes (center in Fig 2B). Thus, at 12 h the four glycoforms behaved more similarly to each other regarding the induction of differential gene expression.

Regarding those genes clasically regulated by FSH in granulosa cells, only after 12 h of exposure to FSH glycoforms was fold increase in the *Lhcgr* and *Cyp191a* genes clearly detectable (S2 Table); in the case of *rec*FSH, expression of *Cyp191a* increased from -0.38 at 6h to 0.87 (log fold increase) at 12h. In contrast, the highest increase in expression of this latter gene after *eq*FSH exposure was observed at 6h (log fold increase, 0.92). Although *Ccnd2* expression could not be detected in our study, one *Ccnd2* transcript (ENSRNOT00000086440) was detectable under the conditions employed; the trend in log fold change identified for this particular transcript after exposure to the pituitary FSH glycoforms, but not to *rec*FSH and *eq*FSH, was of modestly increased values with time (log fold changes at 6h and 12h, respectively: FSH[18/21], from -0.25 to 0.03; FSH[24], from -0.37 to -0.08 at 12h).

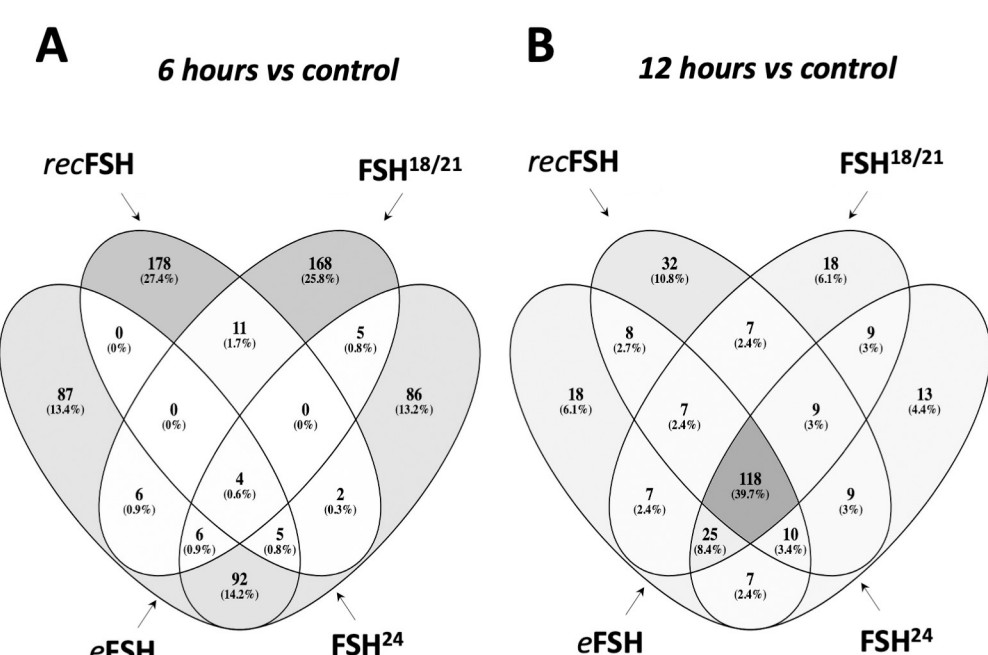

**Fig 2. Venn diagrams of the top-200 differentially expressed genes between FSH glycoforms at 6 h (A) and 12 h (B).** Numbers inside the figure represent the number of genes shared in the corresponding set. Gray scale is proportional to the number of genes in the subset compared to the total of genes. Note that the number of genes shared between the four groups (center of the figure) at 6 h is only 4 genes; meanwhile, at 12 h, the very same set is 118 genes, meaning that at 12 h, the four compounds behave much similar than at 6 h.

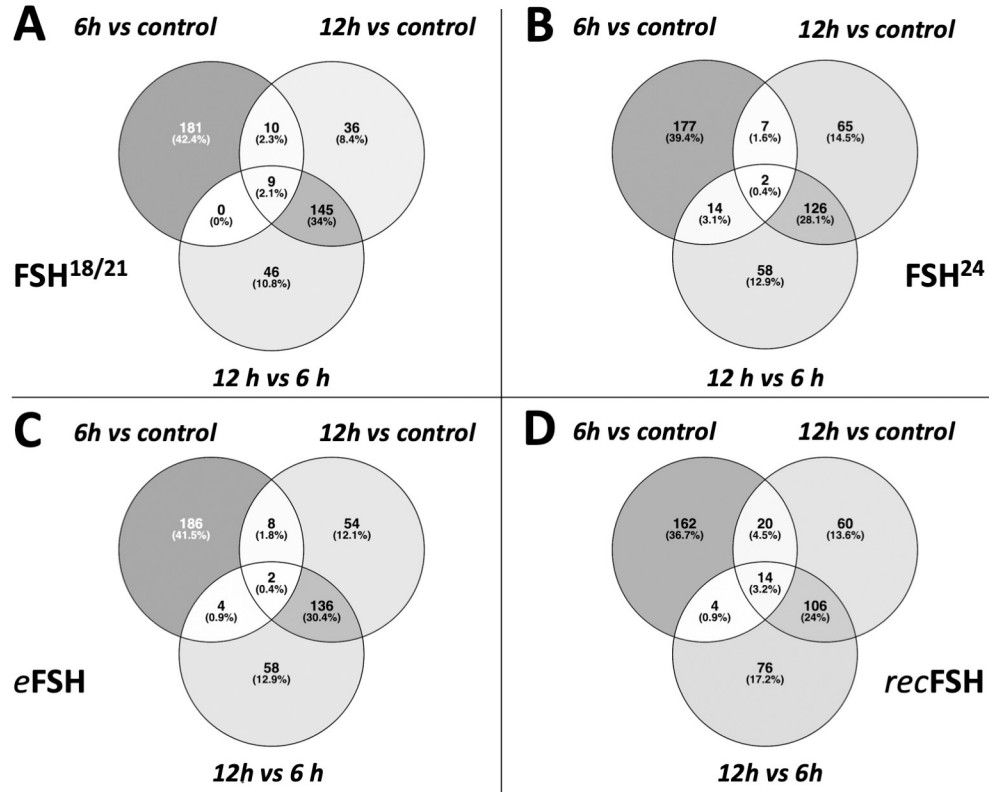

**Fig 3. Contrast between 6 and 12 hours for each FSH glycoform.** In the four cases, the top 200 more differentially expressed genes are more different between 6 hours and control than 12 hours vs control. The contrast of recombinant FSH shows that there is a strong difference at 6 and 12 hours, since the intersection between 12 and 6 hours is only 34 genes.

## Contrasts among each FSH glycoform *vs* control at 6 and 12 hours

Fig 3A–3D shows the contrast between 6 h and 12 h for each FSH glycoform. In all cases, the number of top 200 differentially expressed genes induced at 6 h was higher than at 12 h. *rec*FSH showed the strongest difference between 6 h and 12 h since the intersections between these incubation times were only 34 genes (Fig 3D). The results with all comparisons, as well as the statistics of each contrast can be found at https://osf.io/57j3f/.

## Gene ontology enrichment

Each contrast was enriched by means of the DAVID online tool [51]. Gene Ontology (GO) terms were obtained with the DAVID-standard *p*-values of enrichment (at p<0.01). This method was applied to explore those processes in which differentially expressed genes participate. The four different treatments were analyzed at 6 h and 12 h time points, using the list of overexpressed and underexpressed genes separately.

**Processes associated with overexpressed genes after 6 hours of FSH stimulation (Tables 1 and 2 and Fig 4).** Exposure of granulosa cells to *rec*FSH and FSH[18/21] was followed by induction of several enriched processes that included response to estradiol and calcium, angiogenesis (in the case of *rec*FSH), and response to the gonadotropin stimulus and steroid biosynthetic process (for FSH[18/21]) (Tables 1 and 2). For these two FSH compounds the only category shared was reponse to drugs. In contrast, FSH[18/21] shared several processes with *eq*FSH,

**Table 1. Processes associated with overexpressed genes after exposure for 6 hours to FSH[18/21] and FSH[24].**

| Processes associated with overexpressed genes at 6 hours | | | |
|---|---|---|---|
| FSH[18/21] | | FSH[24] | |
| *Exclusive* | *Associated* | *Exclusive* | *Associated* |
| | | | |
| Cellular response to lipopoly-saccharide<br>Cellular response to gonado-tropin stimulus<br>Cellular response to interferon gamma<br>Response to gonadotropin<br>Cellular response to organic cyclic compound<br>Steroid biosynthetic process | Response to drug<br>Cellular response to cAMP<br>Cellular response to follicle-stimulating hormone stimulus<br>Response to estrogen<br>Response to peptide hormone<br>Ovulation from ovarian follicle<br>Cellular response to epinephrin stimulus | None | Cellular response to cAMP |

including response to estrogen, ovulation, and cAMP response, the latter also shared with FSH[24]. This finding suggests that *eq*FSH and FSH[18/21] behaved more similarly in the early (6 h) response. Interestingly, and in contrast to FSH[18/21], FSH[24] did not exhibit any exclusively enriched process after stimulation during 6 h and showed only one shared biological process (cellular response to cAMP, shared with *eq*FSH and FSH[18/21]). Fig 4 shows the enriched processes of overexpressed genes for the four glycoforms at 6 h, with the nodes corresponding to each glycoform colored in yellow.

**Processes associated with overexpressed genes at 12 hours of FSH stimulation (Tables 3 and 4, and Fig 5).** Similar to the data shown by the Venn diagrams shown in Fig 2, all compounds exhibited more shared processes at 12 h than at 6 h, as illustrated in Fig 5 and Tables 3 and 4. In fact, eleven processes (including response to estrogen and cholesterol biosynthetic process), were shared by all FSH phenotypes. At this time, there were fewer treatment-specific processes than after 6 h of FSH exposure (Tables 1 and 2 and Fig 5), being *rec*FSH and FSH[24] (both tetra-glycosylated FSH molecules) the glycoforms exhibiting a larger number of exclusive processes. FSH[18/21] and *eq*FSH showed only one exclusive process at this time.

**Processes associated with underexpressed genes at 6 hours of FSH stimulation (Tables 5 and 6, and Fig 6).** For the underexpressed genes, there was a lower number of significantly enriched processes (n = 12) despite the gene sets of overexpressed and underexpressed genes were of the same size (*i.e.* 200). FSH[18/21] was the preparation exhibiting more negatively regulated exclusive processes (n = 7), including positive regulation of angiogenesis, whereas for *eq*FSH and *rec*FSH there was only one exclusively enriched process, regulation of cell growth

**Table 2. Processes associated with overexpressed genes after 6 hours of cell exposure to *eq*FSH and *rec*FSH.**

| Processes associated with overexpressed genes at 6 hours | | | |
|---|---|---|---|
| *eq*FSH | | *rec*FSH | |
| *Exclusive* | *Associated* | *Exclusive* | *Associated* |
| | | | |
| Response to cAMP<br>Positive regulation of interleukine-2 production | Cellular response to cAMP<br>Cellular response to follicle-stimulating hormone stimulus<br>Response to estrogen<br>Response to peptide hormone<br>Ovulation from ovarian follicle<br>Cellular response to epinephrin stimulus | Cellular response to interleukin-1<br>Cellular response to calcium ion<br>Cellular response to amino acid stimulus<br>Response to estradiol<br>Response to hypoxia<br>Angiogenesis<br>Microtubule depolymerization<br>Extracellular matriz organization<br>Chromosome segregation<br>Mitotic sister chromatid segregation | Response to drug |

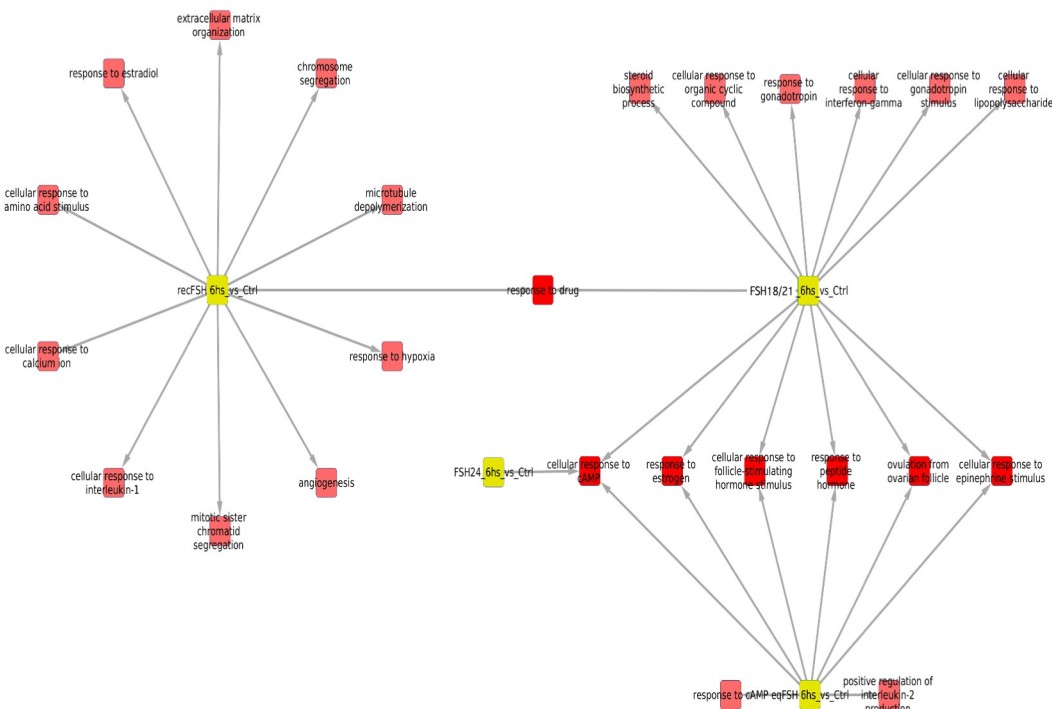

**Fig 4. Enriched processes of overexpressed genes for the four FSH glycoforms at 6 h, with the nodes corresponding to glycoforms in yellow.** In this network representation, red squares represent significantly (p<0.01) enriched biological processes in the treatment in which there is a link. There are some shared processes i.e. biological functions that appear enriched in more than one phenotype, which is indicated by more than one link. It is important to consider the number of associated processes for each isoform: eqFSH, 8 processes (2 exclusive); recFSH, 11 processes (only 1 exclusive); FSH[18/21] 13 processes, from which 6 are exclusive, and FSH[24] only one associated process. In this figure, the intersection between FSH[18/21] at 6 h is depicted against the other 3 isoforms at the same time.

and thyroid hormone metabolic process, respectively. Positive regulation of cell proliferation and cell migration were negatively associated processes between FSH[18/21] and *eq*FSH at this time. For FSH[24] we only identified cell response to cAMP as a negatively regulated shared process (with *rec*FSH).

**Processes associated with underexpressed genes at 12 hours of FSH stimulation (Fig 7 and Tables 7 and 8).** For underexpressed genes, a larger number of significantly enriched biological processes were identified at 12 h than at 6 h (Fig 7). At this time, *eq*FSH was the preparation showing the largest number of significantly underregulated processes (both exclusive and associated) (n = 18 processes *vs* 3 processes at 6 h), from which 9 were shared and 10 were exclusive; the latter were more abundant than those detected with the exclusively overexpressed genes at the same time, during which only 1 exclusive process (cell adhesion) was detected (see Fig 5). In contrast to the findings at 6 h, FSH[18/21] showed fewer shared biological processes at 12 h (n = 4), including DNA replication, chromosome segregation, microtubule-based movement, and response to toxic substances, probably related to decreased cell growth, movement, and proliferation at this time) and no exclusive processes. Meanwhile, exposure to FSH[24] resulted in negative regulation of eight shared processes and five exclusive processes, including enriched inactivation of MAPK activity, bone mineralization, mitosis cytokinesis, and peptidyl-serine phosphorylation. Finally, exposure to *rec*FSH for 12 h yielded 4 shared- and 3 exclusive processes, including cell smooth muscle migration and proliferation, and intracellular signal transduction (Tables 7 and 8).

**Table 3. Processes associated with overexpressed genes after 12 hours exposure to FSH[18/21] and FSH[24].**

| Processes associated with overexpressed genes at 12 hours | | | |
|---|---|---|---|
| FSH[18/21] | | FSH[24] | |
| *Exclusive* | *Associated* | *Exclusive* | *Associated* |
| | | | |
| Phospholipid biosynthetic process | Response to nutrient<br>Steroid biosynthetic process<br>Male gonad development<br>Cellular response to cholesterol<br>Sterol biosynthetic process<br>Extracellular matrix organization<br>Response to hypoxia<br>Response to drug<br>Cholesterol metabolic process<br>Response to estradiol<br>Response to organic cyclic compound<br>Metabolic process<br>Oxidation-reduction process<br>Cholesterol synthetic process<br>Cellular response to follicle-stimulating hormone stimulus<br>Response to estrogen | Cellular response to hypoxia<br>Cellular response to cadmium ion<br>Cellular response to starvation | Male gonad development<br>Isoprenoid biosynthetic process<br>Cellular response to cholesterol<br>Sterol biosynthetic process<br>Extracellular matrix organization<br>Response to hypoxia<br>Response to drug<br>Cholesterol metabolic process<br>Response to estradiol<br>Response to organic cyclic compound<br>Metabolic process<br>Oxidation-reduction process<br>Cholesterol biosynthetic process<br>Cellular response to follicle-stimulating hormone stimuluss<br>Response to estrogen |

**Timeline for each FSH phenotype (S1 to S4 Figs).** *FSH[18/21]*. As shown in S1 Fig, overexpressed gene-associated processes exhibited exclusive biological processes at each time (6 and 12 h), with a subset of shared processes at both times. At 6 h, all FSH[18/21]-stimulated exclusive processes were associated with cell responses, whereas the four shared processes present at 12 h included responses to steroid hormones and drugs, as well as biosynthetic steroid process. At 12 h, 13 processes, including responses to estradiol, cholesterol, hypoxia, nutrient, and organic cyclic compound, as well as male gonad development appeared as significantly enriched. Meanwhile, among underexpressed genes-related processes, positive regulation of

**Table 4. Processes associated with overexpressed genes after 12 hours exposure to *eq*FSH and *rec*FSH.**

| Processes associated with overexpressed genes at 12 hours | | | |
|---|---|---|---|
| *eq*FSH | | *rec*FSH | |
| *Exclusive* | *Associated* | *Exclusive* | *Associated* |
| | | | |
| Cell Adhesion | Cellular response to cAMP<br>Male gonad development<br>Isoprenoid biosynthetic process<br>Extracellular matrix organization<br>Response to hypoxia<br>Response to drug<br>Cholesterol metabolic process<br>Response to estradiol<br>Response to organic cyclic compound<br>Metabolic process<br>Oxidation-reduction process<br>Cholesterol biosynthetic process<br>Cellular response to follicle-stimulating hormone stimulus<br>Response to estrogen | Female gonad development<br>Adrenal gland development<br>Response to ethanol<br>Negative regulation of osteoclast differentiation<br>Response to go-nadotropin<br>Liver development | Cellular response to cAMP<br>Response to nutrient<br>Steroid biosynthetic process<br>Isoprenoid biosynthetic process<br>Cellular response to interleukin-1<br>Cellular response to calcium ion<br>Cellular response to amino acid stimulus<br>Response to estradiol<br>Response to hypoxia<br>Cholesterol metabolic process<br>Response to estradiol<br>Response to organic cyclic compounds<br>Metabolic process<br>Oxidation-reduction process<br>Cholesterol synthetic process<br>Cellular response to follicle-stimulating hormone stimulus<br>Response to estrogen |

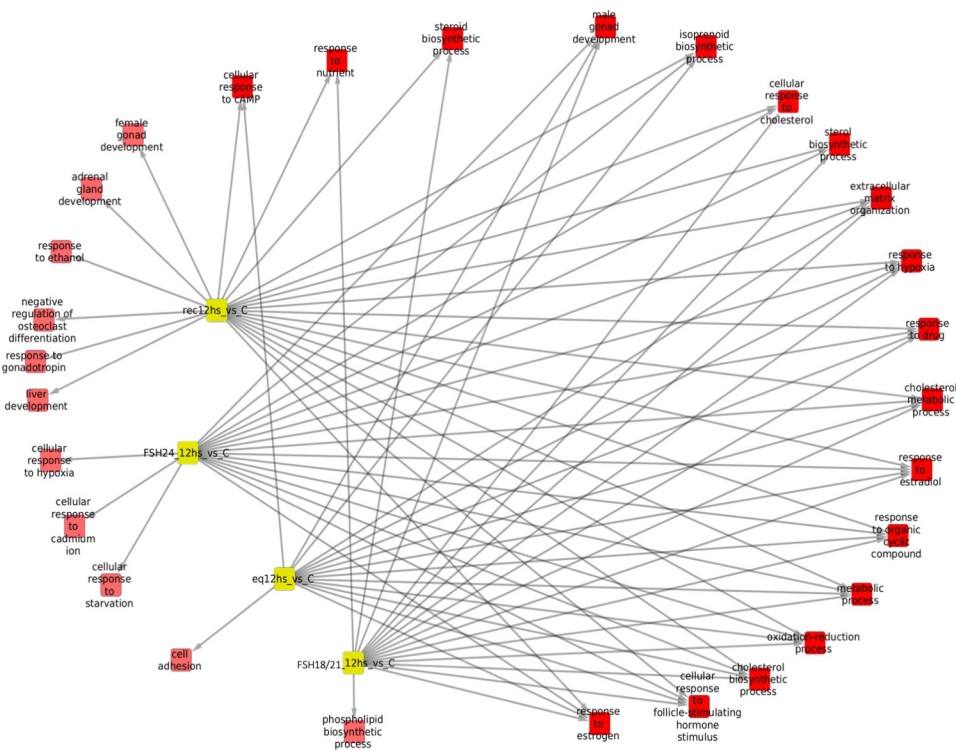

**Fig 5. Biological processes associated with overexpressed genes at 12 hours after FSH glycoforms addition.** Color code is the same as in Fig 4. It is evident that the number of shared processes was larger at 12 h than at 6 h of exposure. See also legend of Fig 4.

proliferation and migration, wound healing, and regulation of gene expression appeared as exclusive at 6 h, and DNA replication, chromosome segregation, microtubule-based movement, and response to substances were enriched at 12 h. Shared underexpressed processes between 6 and 12 h were not evident. Contrary to those corresponding to overexpressed genes, there were fewer underexpressed genes- associated processes at 12h than at 6h.

*FSH[24]*. Interestingly, exposure to FSH[24] for 6 h yielded the same associated process based on overexpressed and underexpressed genes, *i.e.* cellular response to cAMP (S2 Fig bottom). This was the only process enriched at 6 h for this particular compound. On the other hand, several processes associated with cellular response to different stimuli (in the case of

**Table 5. Processes associated with underexpressed genes after 6 hours exposure to FSH[18/21] and FSH[24].**

| Processes associated with underexpressed genes at 6 hours | | | |
|---|---|---|---|
| **FSH[18/21]** | | **FSH[2]** | |
| *Exclusive* | *Associated* | *Exclusive* | *Associated* |
| | | | |
| Response to wounding<br>Positive regulation of gene expression<br>Positive regulation of smooth muscle cell migration<br>Response to lipopolysacch-aride<br>Wound healing<br>Positive regulation of angiog-enesis<br>Positive regulation of smooth muscle cell proliferation | Positive regulation of cell proliferation<br>Positive regulation of cell migration | None | Cellular response to cAMP |

**Table 6. Processes associated with underexpressed genes after 6 hours exposure to *eq*FSH and *rec*FSH.**

| Processes associated with underexpressed genes at 6 hours | | | |
|---|---|---|---|
| *eq*FSH | | *rec*FSH | |
| *Exclusive* | *Associated* | *Exclusive* | *Associated* |
| | | | |
| Regulation of cell growth | Positive regulation of cell proliferation<br>Positive regulation of cell migration | Thyroid hormone metabolic process | Cellular response to cAMP |

overexpressed genes), and cell division, DNA replication, and cell movement (for underexpressed genes), among others, were observed after 12 h of exposure to this glycoform.

*eqFSH*. For overexpressed genes-related processes stimulated after 6 h exposure to *eq*FSH, the response to cAMP and processes associated with response to FSH stimulus (*e.g.* ovulation, response to peptide hormone) were exclusive, whereas response to estrogen and cAMP were shared with those processes stimulated after exposure during 12 h (S3 Fig). Among the exclusive processes detected after 12 h of exposure were cholesterol metabolic and biosynthetic processes, extracellular matrix organization, and cell adhesion. For the case of underexpressed genes-related processes, regulation of cell growth was the only exclusive function, and cell migration and proliferation were shared with processes stimulated at 12 h by this glycoform. Biological processes associated with cell proliferation were among the 12 h exclusive ones. Interestingly enough was finding that the response to estradiol was an enriched process associated with overexpressed and underexpressed genes at 12 h, similar to the cellular response to

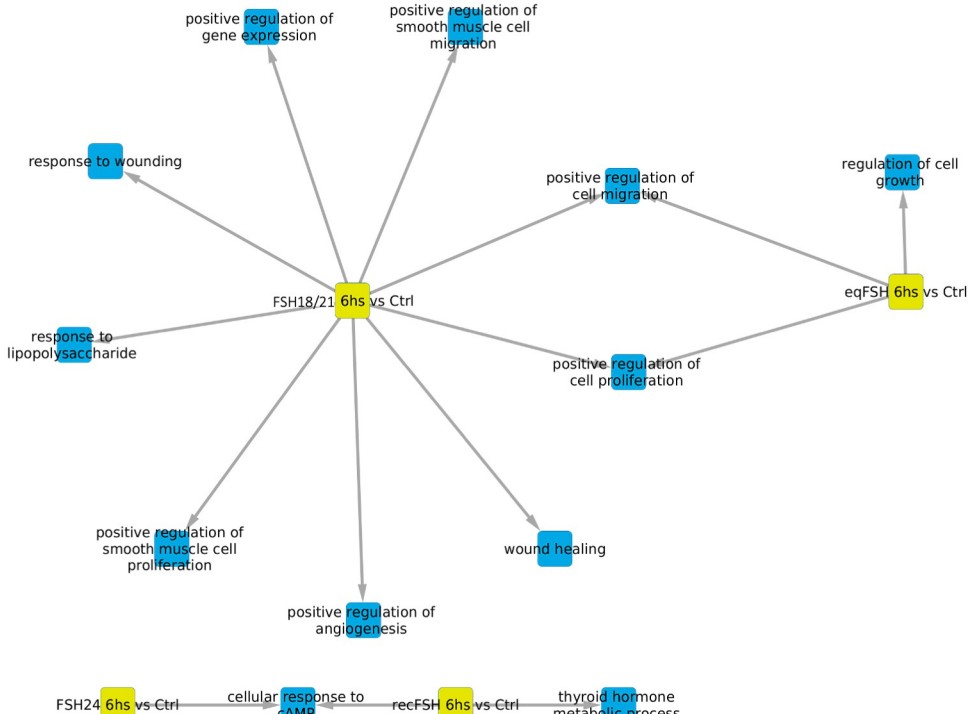

**Fig 6. Biological processes associated with underexpressed genes at 6 hours of exposure to each of the four FSH treatments with the nodes representing each glycoform in yellow.** In this representation, blue color represents the enriched processes.

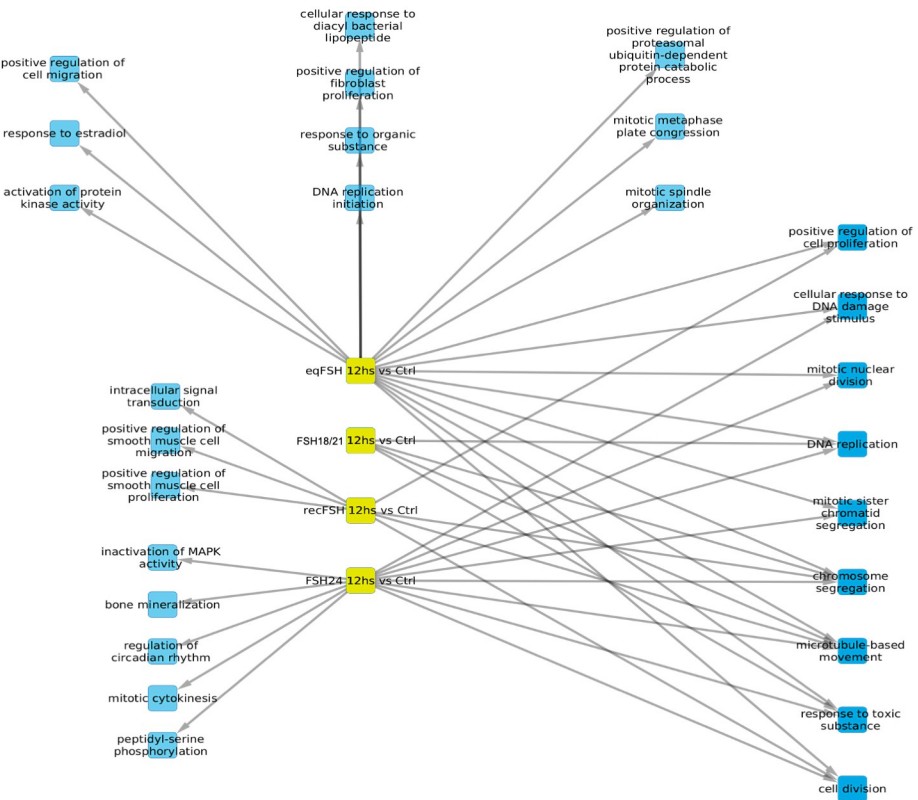

**Fig 7. Biological processes associated with underexpressed genes at 12 hours of exposure to each of the four treatments.** In this representation, blue color represents the enriched processes.

cAMP observed for FSH[24] at 6 h. In both cases, stimulation followed by inhibition of these processes may be related to their selective (biased) desensitization following stimulation with FSH[24] and *eq*FSH.

*recFSH*. At 6 h, microtubule depolymerization, mitotic sister segregation, and angiogenesis were among the exclusive, overexpressed gene-related processes (S4 Fig). Elements shared between 6 h and 12 h were response to drug, hypoxia and estradiol, as well as extracellular

**Table 7. Processes associated with underexpressed genes after 12 hours exposure to FSH[18/21] and FSH[24].**

| Processes associated with underexpressed genes at 12 hours | | | |
|---|---|---|---|
| FSH[18/21] | | FSH[24] | |
| *Exclusive* | *Associated* | *Exclusive* | *Associated* |
| | | | |
| None | DNA replication<br>Chromosome segregation<br>Microtubule-based movement<br>Response to toxic substance | Inactivation of MAPK activity<br>Bone mineralization<br>Regulation of circadian rhythm<br>Mitotic cytokinesis<br>Peptidyl-serine phosphorylation | Cellular response to DNA damage stimulus<br>Mitotic nuclear division<br>DNA replication<br>Mitotic sister chromatid segregation<br>Chromosome segregation<br>Microtubule-based movement<br>Response to toxic substance<br>Cell division |

**Table 8.  Processes associated to underexpressed genes after 12 hours exposure to *eq*FSH and *rec*FSH.**

| Processes associated to underexpressed genes at 12 hours | | | |
|---|---|---|---|
| *eq*FSH | | *rec*FSH | |
| *Exclusive* | *Associated* | *Exclusive* | *Associated* |
| Response to organic substance<br>Mitotic spindle organization<br>DNA replication initiation<br>Cellular response to diacyl bacterial lipopeptide<br>Response to estradiol<br>Positive regulation of fibroblast proliferation<br>Positive regulation of cell migration<br>Mitotic metaphase plate congresssion<br>Activation of protein kinase activity<br>Positive regulation of proteasomal ubiquitin-dependent proteinkinase activity | Positive regulation of cell proliferation<br>Cellular response to DNA damage stimulus<br>Mitotic nuclear division<br>DNA replication<br>Mitotic sister chromatic segregation<br>Chromosome segregation<br>Microtubule-based movement<br>Response to toxic substance<br>Cell division | Intracellular signal transduc-tion<br>Positive regulation of smooth<br>muscle cell migration<br>Positive regulation of smooth<br>muscle cell proliferation | Positive regulation of cell proliferation<br>Chromosome segregation<br>Microtubule-based movement<br>Cell division |

matrix organization. Comparatively, more exclusive processes were detected after 12 h of *rec*FSH stimulation, including female gonad development, steroid biosynthetic process, and response to estrogen, among others.

The only process significantly enriched from underexpressed genes at 6 h was thyroid hormone metabolic process (S4 Fig, bottom left). Microtubule-based movement, cell proliferation and intracellular signaling appeared as processes enriched after 12 h of *rec*FSH exposure. The case of chromosome segregation was interesting in that this process was significantly enriched for overexpressed genes at 6 h and at 12 h for the underexpressed ones, perhaps pointing to a transitional process. Another interesting finding was that cellular response to cAMP appeared as an enriched process related to underexpressed genes at 6 h but also to overexpressed genes at 12 h. This could mean that this particular process changed its behavior from one pole to the other during a 12 h exposure to *rec*FSH.

## Pathway perturbation analysis

Pathway perturbation analyses allow linking the global effect of gene perturbation observed in a given set of experiments with known functional and molecular processes. Based on previous knowledge of the effects of FSH on FSHR-mediated intracellular signaling [29, 30], we focused our analysis on cAMP-PKA, MAPK-, and PI3K/AKT-mediated signaling. Our analyses identified these pathways as altered in virtually all experimental comparisons.

**Behavior of the cAMP signaling pathway in response to each FSH compound (S5 and S6 Figs).**   A closer inspection on the cAMP pathway, allowed us to identify that said dysregulation is driven by different sets of perturbed genes. S5 and S6 Figs show the differentially expressed genes in KEGG-resolved cAMP signaling pathway for each treatment at 6 and 12 h. As shown, each compound exhibited a time-dependent individual molecular signature. For example, 6 h after FSH[18/21] and FSH[24] exposure, membrane-bound FSH-stimulated FSHR expression was found generally underexpressed, most probably due to internalization and subsequent degradation of the receptor. Simultaneously, phosphodiesterase was overexpressed to prevent prolonged activation of the cAMP/PKA pathway. In the case of FSH[18/21], CREB was overexpressed at 6 h, whereas no changes in this transcription factor were observed in the case of FSH[24] at this time. After 6 h of FSH[24] exposure, G protein-coupled receptors (GPCRs) that responded to colinergic and GABAergic stimuli were modestly overexpressed whereas no change was observed for FSH[18/21] at this time. Changes similar to those provoked by FSH[18/21]

exposure for 6 h were observed for *eq*FSH, whereas for *rec*FSH no GPCR exhibited dysregulation.

After 12 h of FSH exposure, the effects of gene expression on pathway molecules were more consistent. Most molecules in the cAMP-PKA pathway (*eg*. CREB) exhibited underexpression at this time point, regardless of their cellular or pathway position. The exception was AMPA (α-amino-3-hydroxy-5-methyl-4-isoxazolepropionic acid) receptor, which is stimulated by glutamic acid present in follicular fluid [52] and in turn regulates signaling molecules (presumably *via* calcium influx) involved in ovulation [53]; AMPAR showed overexpression with all treatments. At this time, signaling was also very similar between FSH[18/21] and *eq*FSH with the exception of GPCRs for glycoprotein hormones and c-Jun N-terminal kinase (JNK; a kinase involved in cell cycle progression, mitosis and apoptosis [54, 55]) signaling overexpression stimulated by *eq*FSH. It is interesting to note the contrast between *rec*FSH- and FSH[24]- (both fully-glycosylated human FSH compounds) stimulated signaling. JNK-mediated signaling and CREB-regulated gene expression (at 6 h) as well as MEK and JNK signaling (at 12 h), were markedly overexpressed after exposure to *rec*FSH but not to FSH[24]. In contrast, after exposure to FSH[24] both CREB-regulated gene expression and JNK signaling remained unmodified or were underexpressed at 6 h and 12 h, respectively, whereas MAPK signaling was slightly overexpressed at both times.

**Behavior of the MAPK and PI3K-AKT signaling pathways in response each FSH compound (S7 to S10 Figs).**   The effects of the FSH compounds tested on the MAPK signaling pathway also differed among the glycoforms (S7 and S8 Figs). Notably, MEK1 and MEK2 pathways were upregulated by FSH[18/21] at 6 h but not by FSH[24], whereas at 12 h, ERK signaling was modestly upregulated in response to both glycoforms. Meanwhile, in the case of *eq*FSH, MEK1 and 2 as well as ERK-signaling were modestly overexpressed at 6 h and 12 h, respectively, whereas after exposure to *rec*FSH, both MEK2 and ERK were overexpressed at 12 h but not at 6 h. Worthy of note in the PI3-AKT signaling pathway (S9 and S10 Figs) was the overexpression of SGK (serum and glucocorticoid-regulated kinase, which is involved in survival signaling, growth, and proliferation [56]) after 12 h of cell exposure to FSH[18/21], *rec*FSH, and *eq*FSH, whereas this signaling pathway remained unaltered or markedly underexpressed at 6 h for all glycoforms.

## Validation of FSH glycoforms-regulated gene expression by RT-PCR

To validate the results obtained from the RNA-seq experiment, we analyzed by RT-PCR the relative mRNA expression of four over-expressed and one under-expressed genes under the stimulation of distinct glycoforms (Fig 8). Genes examined that were putatively up-regulated by FSH glycoforms included: *a)* Phospholipase D1 (*Pld1*), which encodes an enzyme implicated in a number of cellular pathways, including signal transduction mediated by GPCRs and receptor tyrosine kinases, subcellular trafficking, and regulation of mitosis [57]; *b)* Neuropeptide Y (NPY) receptor Y1 (*Npy1R*), which encodes for a GPCR involved in Npy-regulated granulosa cell proliferation and apoptosis [58, 59]; c) Antimüllerian hormone gene (*Amh*), encoding a glycoprotein belonging to the transforming growth factor beta superfamily, which is involved in follicular development [60]; and *d)* Vascular endothelial growth factor B (*Vegf-B*), a gene that encodes the VEGF-B protein, a factor involved in angiogenesis [61]. The FSH down-regulated gene examined was BCL2 like 1 (*Bcl2l1*) which acts as anti- or pro-apoptotic regulators that are involved in a wide variety of cellular activities [62], including prolactin signaling [63]. The selection of those genes for validation was based on the role they play in FSH-stimulated follicular maturation, rather than on fold change variation, and their corresponding mRNAs were quantified in three independent biological replicates (samples coming from

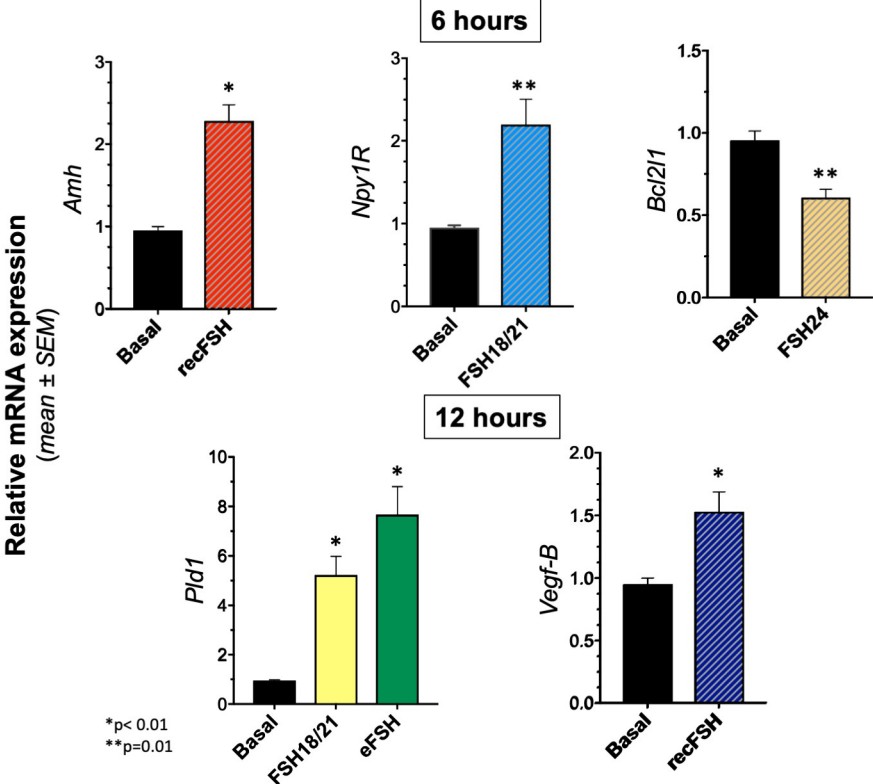

**Fig 8. Validation of differentially expressed genes in the RNA-seq data by RT-PCR of five selected FSH glycoform-sensitive genes (*Pld1*, *Npy1R*, *Amh*, *Vegf-B*, and *Bcl2l1*).** Genes examined that were putatively up-regulated by FSH glycoforms included: phospholipase D1 (*Pld1*); neuropeptide Y (NPY) receptor Y1 (*Npy1R*), antimüllerian hormone gene (*Amh*), and vascular endothelial growth factor B (*Vegf-B*), whereas the FSH down-regulated gene examined was BCL2 like 1 (*Bcl2l1*).

three different culture wells) from granulosa cells exposed to different FSH glycoforms during 6h or 12h.

The mRNA expression of the 4 up-regulated genes in response to particular glycoforms *vs* zero hours [with statistically significant (p<0.01 or p<0.05) differences] was the following: *Amh* and *Npy1R* for *rec*FSH and FSH[18/21] at 6 h, respectively; *Pld1* for FSH[18/21] and *eqFSH* at 12 h; and *Vegf-B* for *rec*FSH at 12 h. For the down-regulated gene (*Bcl2l1*) a statistically significant difference (p<0.05) was found for FSH[24] at 6 h. These results indicated the existence of a positive validation (Fig 8). The use of only 3 replicates was justified by the difficulties in generating more material from such experiments. It is clear that using more replicates may better support the PCR *vs* RNAseq results where statistical power is increased by the measurement of thousand of targets at the same time.

## Discussion

Previous studies have demonstrated that hypoglycosylated FSH (FSH[18/21]) is the predominant form secreted by women in reproductive age and that a shift to fully-glycosylated glycoforms occurs as age advances, which is probably due to the progressive decrease in the production of ovarian estrogens as menopause approaches [15, 64–66]. *In vitro* and *in vivo* studies also have shown that hypoglycoslated FSH exhibits higher FSHR binding activity and receptor affinity

as well as potency and efficiency to trigger intracellular signaling than fully glycosylated FSH [12, 14, 15, 26, 27, 65, 67]) despite its relatively shorter plasma half-life [67, 68]. In fact, when injected to *Fshb* null female mice, both hypo- and fully glycosylated FSH stimulated ovarian weight response and induced particular ovarian genes in a similar fashion, whereas in *Fshb* null male mice, hypoglycosylated FSH[18/21] was more active than fully-glycosylated FSH[24] in inducing FSH-responsive genes and Sertoli cell proliferation [27]. More recently, short-term *in vivo* studies in prepubertal mice showed that FSH[18/21] was more effective than FSH[24] in promoting follicle development and health, as well as in rescuing granulosa cells from apoptosis [67]. Although ovarian gene transcription *in vivo* was stimulated by both glycoforms, FSH[18/21] induced greater activation of Gαs-mediated cAMP-PKA signaling as well as PI3/K and MAPK/ERK signaling pathways, with greater expression of early response genes upon administration of this particular glycoform [67]. These data were concordant with findings from previous studies showing that *in vivo* administration of more basic, short-lived FSH isoforms to hypophysectomized rats, mantained equally or even more efficiently granulosa cell proliferation than their more acidic counterparts [22].

Of note, it was interesting to find that mainly after 12h exposure to the FSH glycoforms, was fold increase in *Lhcgr* (and also *Cyp19a1*, in the case of the pituitary FSH glycoforms) clearly detectable. This may be explained by the fact that estradiol production (and also presumably *Cyp19a1* mRNA expression) in culture conditions similar to that employed in the present study occurs late during the incubation period (24–48 h) [40, 69]. A similar expression profile has also been observed for the *Lhcgr* [70, 71]; the finding that mRNA expression of the *Lhcgr* occurs quite late in cultured granulosa cells (peak levels at 48 h) [71], may explain the relatively modest increase in the transcript of this particular gene after 12 h of FSH exposure. *eq*FSH and *rec*FSH genes also exhibited a strong expression of the *Lhcgr* at 12 h, but not of that of *Cyp19a1*. This latter finding may suggest an even slower expression of this particular gene when stimulated by these particular glycoforms, which could not be detected with the time resolution of the present study. The same occurred in the case of the *Ccnd2* gene, underlying the need to perform experiments with a more frequent time resolution. A complete explanation to this latter finding remains to be established. Nevertheless, either differences in culture conditions between ours and a previous study on *Ccnd2* mRNA expression [72] or a low time resolution for clearly detect expression of this particular gene in our study, might explain these apparent discrepancies.

In the present study, we analyzed whether FSH glycoforms exhibiting distinct glycosylation patterns differentially induce gene transcription and activation of biological processes and signaling pathways in cultured rat granulosa cells exposed during 6h and 12h to equal doses of the glycoforms. Analysis of differentially expressed genes identified particular changes in gene expression induced by each glycoform, with a limited number of genes *shared* among the glycoforms at 6 h, followed by a sharp increase at 12 h, time at which the four glycoforms behaved more similarly for inducing differential gene expression. These data reflect the relatively early differential effects of the glycoforms on gene expression, with a platau reached at 12 hours. Analysis of individual FSH glycoforms revealed the expression of more FSH-dependent genes at 6 h than at 12 hours, particularly for the hypo-glycosylated preparations (FSH[18/21] and *eq*FSH), which was expected given that FSH-mediated signaling occurs rapidly after receptor activation in *in vitro* conditions and that hypoglycosylated FSH exhibits a more favorable FSHR binding profile than its fully-glycosylated counterpart from kinetic and thermodynamic points of view, as disclosed by molecular dynamics simulations [12, 28]. Nevertheless, the data also showed that even after 12 h of FSH exposure the effects of FSH stimulation on gene expression persisted, independently of the kinetics by which each glycosylation variant activates the FSHR and stimulates intracellular signaling.

Gene ontology analysis revealed that exposure of granulosa cells to each FSH compound during 6 h or 12 h was differentially associated with induction or inhibition of one or several genes linked *to distinct* biological processes. Some of these processes were exclusive or unique for a given FSH compound whereas others where shared by or associated with several FSH glycoforms, demonstrating the distinctly different ability of each glycoform to activate/inhibit FSH-dependent biological processes. In this analysis, it was interesting to find that in contrast to hypo-glycosylated and *rec*FSH, stimulation with FSH[24] and *eq*FSH for 6 h, was poorly associated with exclusive overexpressed processes. In fact, in the case of FSH[24] no exclusive biological processes and only one associated process (cAMP response) were clearly identified at this time. This observation contrasted with the findings after 12 hours of FSH exposure, where multiple associated processes where shared among all glycoforms, again emphasizing the long-term effects of FSH on biological effects *in vitro*. Here it is important to emphasize that at this time (12 h) the exclusive processes stimulated by FSH[18/21] where virtually absent, underlying the short and acute effects of this particular glycoform on granulosa cell responsiveness, an observation that is in agreement with a previous study on the effects of FSH[18/21] and FSH[24] on ovarian global trascriptomics *in vivo* [67]. In this scenario it is possible to conclude that *rec*FSH (tetra-glycosylated), was the most potent preparation to evoke particular/exclusive biological processes at 6 h and 12 h and that the effects of all FSH compounds converged in a number of similar processes in a time-dependent fashion, as it probably occurs *in vivo* during follicular maturation.

Meanwhile, individual differences in time also were observed in processes associated to underegulated genes, with exposure to FSH[18/21] for 6 h resulting in more underegulated genes that in those by other glycoforms but not later, at 12 h, time at which FSH[18/21] did not show any exclusive process, an observation that was in sharp contrast with the effects of the remaining glycoforms. These data indicates that different glycoforms also have distinct abilities to turn off selective genes in a time dependent manner, with the FSH[18/21] hypo-glycosylated glycoform acting faster than the other compounds. This observation is in line with the ability of this particular glycoform to efficiently trigger intracellular signaling and also to rapidly evoke down-regulation of stimulated signaling and also probaby resensitization of the FSH[18/21]/FSHR complex [12, 14, 26]. In fact, it has been shown that FSH glycosylation can influence the turn-on time of FSHR-mediated signaling [26, 73, 74].

Pathway perturbation analysis unveiled interesting differential effects of the FSH glycoforms on intracellular FSH-regulated signaling. We analyzed by this bioinformatics procedure three well-known pathways stimulated by gonadotropins, the canonical cAMP-PKA as well as the MAPK and PI3K-AKT signaling pathways [29, 30, 75]. It is believed that these signaling cascades lead to fine-tuning regulation of the FSH stimulus, where activation and/or inhibition of their downstream activated components vary depending on the cell context, cell developmental stage, and concentration of the ligand and the receptor. Simultaneous activation of these signaling modules eventually converges on the ligand/receptor-integrated biological responses, which include cell proliferation, differentiation and survival of responsive cells and, at the molecular level, differential gene expression, as observed in the present study. In the cAMP pathway, it was interesting to find that for FSH[18/21] but not for the fully glycosylated FSH[24], the transcription factor CREB was overactivated at 6 hours indicating a rapid activation of the cAMP-PKA pathway by this particular naturally occurring glycoform, whereas after 12 h stimulation of this pathway persisted active for both glycoforms, albeit to a lesser extent after FSH[24] exposure. The overall data underline the differential activation of the cAMP signaling pathway by these two glycoforms.

The MAPK and PI3K-AKT cascades are also part of the intertwined signals regulated by the FSH/FSHR complex and both are involved in controlling follicle development as well as in

regulating gene transcription [29, 76–78]. In these pathways, the effects of the FSH compounds tested on the MAPK signaling pathway also exhibited time-related differences. For example, MEK1 and MEK2 pathways were upregulated by FSH[18/21] at 6 h *but not* by FSH[24], whereas at 12 h, ERK signaling was modestly upregulated in response to both glycoforms. Worthy of note is the fact that activation of FSHR involves cross-talk with pathways regulated by growth factors and the estrogen GPCR (GPER) [56, 79, 80]). In the former, RTK plays an important role in folliculogenesis and ovulation [81, 82], and it is known that this kinase may activate JNK in a Src and PI3K dependent fashion, whereas in the latter FSHR/GPER heterodimers trigger anti-apoptotic/proliferative signaling through the Gβγ dimer [79, 83]. In this vein, it was interesting to find that all FSH glycoforms activated RTK and MEK after 6 h of FSH exposure, but not later, underlying the important role of this pathways on FSH-evoked cell proliferation and angiogenesis.

Regarding all these signaling pathway analyses, it is important to emphasize on that pathway perturbation captures a *static representation* of what is essentially a dynamic process involving several signaling cascades regulated by distinct factors either directly by the FSHR or via cross-talk with other membrane receptors. As such, unexpected behaviors such as under- and overexpression of certain molecules can be capturing dynamic feedback effects. With this in mind, the differences and similarities observed between different pathways can be thought to be capturing different kinetic behaviors by the distinct FSH treatments. In any case, these data underline the distinct time- and glycosylation-dependent fingerprints for each particular FSH compound.

In summary, the present transcriptomic analysis allowed dissection of some distinctly different glycosylation-dependent effects of the human FSH glycoforms after exposure of cultured rat granulosa to these compounds at different times, comparing these effects with those of *eq*FSH, which albeit differing in amino acid sequence and glycosylation pattern, exerts potent effects at the human FSHR [26]. Of particular interest are the data on the transcriptomic effects of the naturally-occurring FSH glycoforms, which are consistent with previous *in vivo* and *in vitro* studies showing that hypoglycosylated FSH[18/21] exhibits greater biological activity at the target cell level and that both FSH[21/18] and FSH[24] initiate different FSHR-mediated intracellular signaling activation, eventually leading to differential impacts on follicle development [67, 84]. The overall findings are important to better understand the physiological significance of FSH glycoforms in follicular maturation and ovulation in naturally-occurring cycles in women.

## Supporting information

**S1 Table. Representative subset of overexpressed genes at 6 hours of FSH glycoform exposure.**
(PDF)

**S2 Table. Representative subset of overexpressed genes at 12 hours of FSH glycoform exposure.**
(PDF)

**S3 Table. Representative subset of underexpressed genes at 6 hours of FSH glycoform exposure.**
(PDF)

**S4 Table. Representative subset of underexpressed genes at 12 hours of FSH glycoform exposure.**
(PDF)

**S1 Fig. Enriched processes for FSH18/21 at 6 and 12 hours.** In this representation, the red color corresponds to overexpressed enriched processes and the blue color to underexpressed processes. The intersection of the overexpressed processes at 6 h is against the underexpressed processes at 12 h, that is, the comparasion is performed between the two incubation times. This is why the exclusive processes may be different than those shown in Fig 4.
(PDF)

**S2 Fig. Enriched processes for FSH24 at 6 and 12 hours.** In this representation, the red colored squares corresponds to overexpressed enriched processes and the blue color to underexpressed processes.
(PDF)

**S3 Fig. Enriched processes for eqFSH at 6 and 12 hours.** The red squares correspond to overexpressed enriched processes and the blue color to underexpressed processes.
(PDF)

**S4 Fig. Enriched processes for recFSH at 6 and 12 hours.** The red squares correspond to overexpressed enriched processes and the blue color to underexpressed processes.
(PDF)

**S5 Fig. Cyclic AMP signaling pathway (KEGG hsa04024), with differentially expressed genes colored.** This pathway is significantly perturbed by both FSH18/21 and FSH24 at 6h. Note the activation of CREB followed by an inhibition of c-fos induced by FSH18/21, not observed with FSH24. In eqFSH and recFSH, this pathway is significantly perturbed at 6 h. Note the difference in phosphodiesterase (PDE) states: induced by eFSH, and repressed by recFSH. [Pathway perturbation detected with GAGE. Pathway visualization rendered with Pathview].
(PDF)

**S6 Fig. cAMP signaling pathway (KEGG hsa04024) at 12 h, with differentially expressed genes colored.** At this time, this pathway was significantly perturbed by both FSH18/21 and FSH24. Note the activation of PAR1 (coagulation factor II thrombin receptor) by FSH24, which was not induced by FSH18/21. This pathway was significantly perturbed by both recFSHand eFSH at this time. In addition, note the activation of adenylate cyclase (AC) and lipase E (HSL) by recFSH, not induced by eFSH. [Pathway perturbation detected with GAGE. Pathway visualization rendered with Pathview].
(PDF)

**S7 Fig. Within the context of the MAPK signaling pathway (KEGG hsa04010), it can be observed distinct effects of the different FSH glycoforms at the 6-hour time point.** Notably, receptor signaling proteins presented varying expression levels: FSH24 induced a more profound overexpression than FSH18/21 and eqFSH, while recFSH elicited comparatively lower expression. Furthermore, our analysis revealed that highly central molecules within the MAPK pathway, such as p38, responded differentially to the FSH glycoforms. Under the influence of FSH24, p38 and similar central molecules exhibited underexpression. In contrast, when exposed to all other FSH glycoforms, these central molecules (mainly p38) exhibited a consistent pattern of overexpression. [Pathway perturbation was detected using GAGE, and the pathway visualization was generated through Pathview].
(PDF)

**S8 Fig. In the context of the MAPK signaling pathway (KEGG hsa04010), the comparison of the effects among the FSH glycoforms after 12-hour exposure with those at 6 h revealed**

**significant differences.** Receptor signaling proteins are now predominantly underexpressed, regardless of the particular FSH treatment applied, with a marked shift from the earlier (6 h) overexpression. Notably, at this time central molecules, like p38, consistently exhibited overexpression across all FSH glycoforms perturbations. [Pathway perturbation was detected using GAGE, and the pathway visualization was generated through Pathview].
(PDF)

**S9 Fig. Distinct findings emerged in the PI3K signaling pathway (KEGG hsa04151) at the 6 h time point.** Receptor signaling proteins expression showed a balanced distribution of both up- and down regulation across all FSH glycoforms, with no significant differences. PI3K consistently exhibited underexpression across all glycoforms, indicating a shared regulation. Notably, downstream effectors within the PI3K pathway showed unique responses; recFSH upregulated effectors such as CCND1, CDK, and cyclin; and eqFSH, FSH18/21 and FSH24 downregulated BCl-2 and c-Myb. [Pathway perturbation was detected using GAGE, and the pathway visualization was generated through Pathview].
(PDF)

**S10 Fig. In the PI3K pathway (KEGG hsa04151) at the 12 h mark, different features were detected.** First, expression of signaling molecules exhibited a more varied pattern which was distinctly influenced by each FSH glycoform, suggesting differential regulatory effects. For instance, in the case of the proteins involved in focal adhesion, ECM, ITGA, and ITGB were, respectively: a) all upregulated by recFSH; b) up-, down-, and down-regulated, respectively, by eqFSH; and c) down-, up-, and down-regulated, respectively by FSH18/21 and FSH24. In the case of downstream effectors, the patterns of expression of molecules such as GYS and PEPCK were unique for each glycoform: a) both upregulated by recFSH; b) down- and upregulated, respectively, by eqFSH; c) down- and up-regulated, respectively, by FSH18/21,; and c) up-regulated and without change, respectively, by FSH24. All glycoforms consistently showed underexpression of PI3K persisted across all compounds, indicating a sustained (and shared) impact on this key element. [Pathway perturbation was detected using GAGE, and the pathway visualization was generated through Pathview].
(PDF)

## Acknowledgments

The authors would like to thank Dr. James A. Dias, from the State University of New York, Albany, NY, USA, for his carefull reading and comments on this study. Dr. Viktor Y. Butnev passed away before the submission of the final version of this manuscript. Dr. Alfredo Ulloa-Aguirre accepts responsibility for the integrity and validity of the data collected and analyzed.

## Author Contributions

**Conceptualization:** Alfredo Ulloa-Aguirre.

**Data curation:** Georgina Hernández-Montes.

**Formal analysis:** Jesús Espinal-Enriquez, Guillermo De Anda-Jáuregui, Georgina Hernández-Montes, Enrique Hernández-Lemus, Alfredo Ulloa-Aguirre.

**Funding acquisition:** Alfredo Ulloa-Aguirre.

**Investigation:** Teresa Zariñán, Saúl Lira-Albarrán, Rosa G. Rebollar-Vega, George R. Bousfield, Viktor Y. Butnev, Alfredo Ulloa-Aguirre.

**Methodology:** Teresa Zariñán, Guillermo De Anda-Jáuregui, George R. Bousfield, Viktor Y. Butnev, Enrique Hernández-Lemus, Alfredo Ulloa-Aguirre.

**Project administration:** Alfredo Ulloa-Aguirre.

**Resources:** Jesús Espinal-Enriquez, Guillermo De Anda-Jáuregui, Alfredo Ulloa-Aguirre.

**Supervision:** Alfredo Ulloa-Aguirre.

**Validation:** Rubén Gutiérrez-Sagal, Alfredo Ulloa-Aguirre.

**Visualization:** Guillermo De Anda-Jáuregui, George R. Bousfield.

**Writing – original draft:** Guillermo De Anda-Jáuregui, Alfredo Ulloa-Aguirre.

**Writing – review & editing:** Jesús Espinal-Enriquez, George R. Bousfield, Enrique Hernández-Lemus, Alfredo Ulloa-Aguirre.

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
