## [Decision Letter · Decision Letter 0]

2 Feb 2024

PONE-D-23-33249Differential effects of follicle-stimulating hormone glycoforms on the transcriptome profile of cultured rat granulosa cells as disclosed by RNA-seqPLOS ONE

Dear Dr. Ulloa-Aguirre,

Thank you for submitting your manuscript to PLOS ONE. After careful consideration, we feel that it has merit but does not fully meet PLOS ONE’s publication criteria as it currently stands. Therefore, we invite you to submit a revised version of the manuscript that addresses the points raised during the review process.

Please submit your revised manuscript by  Mar 18 2024 11:59PM. If you will need more time than this to complete your revisions, please reply to this message or contact the journal office at plosone@plos.org. Please include the following items when submitting your revised manuscript:

We look forward to receiving your revised manuscript.

Kind regards,

Satish Rojekar, Ph.D.

Academic Editor

PLOS ONE

 [CONACyT, Mexico (grant no. 240619)].  

[This study was supported by grants from CONACyT, Mexico (grant no. 240619) and the Coordinación de la Investigación Científica-UNAM, Mexico (to A.U-A). G.R.B. and VYB were supported by NIH grant P01AG-029531.]

 [CONACyT, Mexico (grant no. 240619)]

6. Please ensure that you include a title page within your main document. You should list all authors and all affiliations as per our author instructions and clearly indicate the corresponding author.

Reviewers' comments:

Reviewer's Responses to Questions

**Comments to the Author**

1. Is the manuscript technically sound, and do the data support the conclusions?

Reviewer #1: No

Reviewer #2: Yes

2. Has the statistical analysis been performed appropriately and rigorously? 

Reviewer #1: No

Reviewer #2: Yes

3. Have the authors made all data underlying the findings in their manuscript fully available?

Reviewer #1: Yes

Reviewer #2: Yes

4. Is the manuscript presented in an intelligible fashion and written in standard English?

Reviewer #1: Yes

Reviewer #2: Yes

5. Review Comments to the Author

Reviewer #1: There are multiple concerns that the authors need to address.

Lines 48-50: It is easier for readers to understand if the FSH glycoforms are numbered. Though it is clear in Methods that four glycoforms were used, with the use of “and” and commas in this sentence it is not obvious how many different glycoforms, that is groups of treatment, are used in this study.

Lines 129-130: FSH is glycosylated at 2 Ns in alpha and beta chains each. Why is FSH-beta is identified as tetra- and tri-glycosylated? Shouldn’t it be mono- and di-glycosylated FSH-beta?

Line 166: diverssignaling should be changed to “diverse signaling”.

Line 174: It should be FSH18/21

Line 199-209: How many rats were used? How many granulosa cells were obtained from each rat? Were granulosa cells from all rats pooled before seeding to plates? Granulosa cells were seeded at 1M per well and cultured for 48h before FSH treatment. Did this culture increase the cell number?

Line 213: What is triplicate incubation? Was each FSH prep for each time-point added to three wells? Was each biological replicate run on different day? How were technical and biological replications considered?

Line 227-228: If only samples with >8 RIN were used, how many samples per treatment per time-point reached this quality threshold? With only 3 replicates, how many ended up being sequenced for each group? How was statistical analysis considered if the number was too low?

Line 231: 10-15 million reads doesn’t appear to be deep. What is the rationale for this depth?

Line 237: how did authors test if the biological processes were indeed active? Mere enrichment of a process among DEGs doesn’t mean active.

Lines 320-350: These two sections are confusing and not well justified. Why did authors run these comparisons? What physiological relevance do these comparisons address? While the first section title suggests “contrast between 6 and 12h”, the text seems to describe contrasts among FSH glycoforms at each time-point! The second section title suggests contrasts among each FSH glycoforms, but the text seems to describe how one glycoform is different from others.

Lines 357-358: It is not recommended to separate up and downregulated genes to discover enriched BPs and pathways as some of the genes involved in a particular pathway may be upregulated while other of that pathway may be down regulated (as in Lines 514-515).

Lines 496-556 : Each of the FSH glycoform appears to have regulated dramatically different set of biological processes. FSH18/21 induced steroid biosynthetic process; FSH24 did not but response to cAMP was both up and down-regulated; eqFSH stimulated processes like ovulation as well as ECM organization that is mostly associated with ovulation; recFSH induced angiogenesis and ECM remodeling processes. Of these, only FSH18/21 appears to have regulated biologically relevant process for follicular granulosa cells. Therefore, it is important for authors to first check which glycoform regulates the usual suspects of granulosa cell genes (Cyp19, Lhcgr, Ccnd2) first before starting to compare different glycoforms.

Authors should first prove that all FSH treatment stimulated Cyp19a1 and Lhcgr in granulosa cells, to ensure that results obtained using the cell culture model of this study are relevant to in vivo biology.

Reviewer #2: In this study the authors have addressed an important question has to how variations in glycosylation of glycoprotein hormones may impart distinct biological features which is largely ignored on studies on signaling effects of hormones on the target tissues. The paper is potential useful because glycosylation variants of hormone is produced in pituitary and which in turn can vary with age leading to possible differential effects. The approach they have used to get to this problem is by a doing global RNA seq analysis on rat granulosa cells after treating them with four different glycoforms of FSH. Furthermore, they relate the differentially expressed genes to intracellular signaling by carrying extensive pathway analysis. The study is very strong in RNA seq analysis and validation of the key genes that are differentially expressed and at times rather heavy technically. The methodology in the paper is well detailed for replication of the study. The results are supported by figures that are clear although with rather wordy legends. The paper is discussed well and futher supported by relevant references. However, I must say that manuscript is not organized well because the figure legends and tables that describe are embedded within text which makes it hard for the reader. Overall the paper is a significant contribution to our understanding of glycoprotein variation and their physiological effects. I have the following comments

1) The author’s observer a rather robust change in the number of the differentially expressed genes between 6hrs and 12hrs. What is the logic in choosing the 6 and 12hrs time period? How is the effect of non-synchronized target cells ruled as cause of this difference?

2) It not clear in the discussion as to what is cause of these changed glycoforms with age?

3) Do the different glycoform have similar affinity to the cognate receptor? Affinity measurements data would have clarified it.

6. PLOS authors have the option to publish the peer review history of their article (what does this mean?). If published, this will include your full peer review and any attached files.

Reviewer #1: No

Reviewer #2: **Yes: **Rauf Latif

---

## [Author Response · Author response to Decision Letter 0]

27 Feb 2024

Editor comments: 

https://journals.plos.org/plosone/s/file?id=ba62/PLOSOne_formatting_sample_title_authors_affiliations.pdf. Answer: Checked

2. To comply with PLOS ONE submissions requirements, in your Methods section, please provide additional information regarding the experiments involving animals and ensure you have included details on (1) methods of sacrifice, (2) methods of anesthesia and/or analgesia, and (3) efforts to alleviate suffering.Answer: INFORMATION IS NOW PROVIDED IN MATERIALS AND METHODS LINES 207-209.

 [CONACyT, Mexico (grant no. 240619)]. Answer: finantial disclosure is now in the cover letter, as the publisher dos not allow to change the initial disclosure.

If this statement is not correct you must amend it as needed. Answer: Amended

Please include this amended Role of Funder statement in your cover letter; we will change the online submission form on your behalf. Answer, done.

[This study was supported by grants from CONACyT, Mexico (grant no. 240619) and the Coordinación de la Investigación Científica-UNAM, Mexico (to A.U-A). G.R.B. and VYB were supported by NIH grant P01AG-029531.] Answer: this is how it should appear.

 [CONACyT, Mexico (grant no. 240619)]. Answer: It should be: CONACyT, Mexico (grant no. 240619) and t de la Investigación Científica-UNAM, Mexico (to A.U-A). G.R.B. and VYB were supported by NIH grant P01AG-029531." 

Please include your amended statements within your cover letter; we will change the online submission form on your behalf. Done

Authors’ responses to reviewers’ comments

PONE-D-23-33249

Differential effects of follicle-stimulating hormone glycoforms on the transcriptome profile of cultured rat granulosa cells as disclosed by RNA-seq

Review Comments/Answers to/from the Author

Reviewer #1: There are multiple concerns that the authors need to address.

- Lines 48-50: It is easier for readers to understand if the FSH glycoforms are numbered. Though it is clear in Methods that four glycoforms were used, with the use of “and” and commas in this sentence it is not obvious how many different glycoforms, that is groups of treatment, are used in this study. 

Answer: 

Thanks for this observation. We have employed the accepted nomeclature familiar to those interested in the gonadotropin field. All glycoforms from the pituitary are known as FSH24 and FSH18/21, whereas the abbreviations of recFSH and equine FSH are also well-known. The identification of the compounds with letters, as suggested by the reviewer is now included in the Abstract (lines 49-51) and Introduction (lines 178-181) of the new ms version. This undoubtedly will facilitate the readers to identify the preparations studied. 

- Lines 129-130: FSH is glycosylated at 2 Ns in alpha and beta chains each. Why is FSH-beta is identified as tetra- and tri-glycosylated? Shouldn’t it be mono- and di-glycosylated FSH-beta? 

Answer:

Reviewer is right on this observation. However, for the identification of tetra- and tri-glycosylated, the heterodimer is taken into account, rather than the isolated subunits. Text has been modified to make clearer and less confusing this issue (lines 129-138 in new version).

- Line 166: diverssignaling should be changed to “diverse signaling”.

Answer:

“divers signaling” has been corrected in the revised version.

- Line 174: It should be FSH18/21

Answer:

This mistake has been corrected in the revised version. Thanks to the reviewer for picking this typo. 

- Line 199-209: How many rats were used? How many granulosa cells were obtained from each rat? Were granulosa cells from all rats pooled before seeding to plates? Did this culture increase the cell number?

Answer:

A total of 22 rats were used to obtain the granulosa cells. We did not count for granulosa cell number from each follicle/rat as after puncturing the follicles cells were were pooled, counted for viable cells and then seeded at 1 million per well and cultured for 48h before FSH treatment. We did not control for changes in cell number, as it was assumed that changes in this variable, if any, would be minimal, and would be equal for all incubations, particularly because pre-incubations and incubations were done in the absence of mitogenic factors (eg. insulin and serum) as well as estrogens. On the other hand, cells were immediately frozen after adding Trizol reagent to prevent any RNA degradation, so there was no available material to perform cell counting at the end of each incubation period (0, 6h and 12h, for 4 FSH preparations at each time). The authors recognize that although a previous study detected changes in the number of cultured granulosa cells (Han et al., Biol Reprod 88 (3):57, 2013; Liu et al., Mol Cell Endocrinol 21:63, 1981, see below), others have not even controlled for this potential effect (see Jia and Hsueh, Endocrinology 119(4):1580, 1976; Shi and Segaloff, Mol Endocrinol 9:734, 1995). The only study we are aware that FSH-stimulated DES-primed granulosa cell proliferation was Wan Shum's (Liu et al., Mol Cell Endocrinol 21:63, 1981) FSH assay using these cells in M199 with 10% chicken serum without 0.5 µM testosterone; growth was only observed after 5-6 days. 

-Line 213: What is triplicate incubation? Was each FSH prep for each time-point added to three wells ? Was each biological replicate run on different day? How were technical and biological replications considered? 

Answer:

Cells were exposed to the different FSH preparations in triplicate wells for each compound and incubation time (including control incubations in the absence of FSH) (ie. for each time point, each FSH preparation was added to three independent wells). For all glycoforms and experimental conditions, the complete set of biological replicates were included in the same NGS run, and all biological replicates were considered for the analysis. Technical replicates were not necessary because the technique employed (NGS) is considered quite robust and sensitive, as per the recommendation of the manufacturer (Illumina). 

- Line 227-228: If only samples with >8 RIN were used, how many samples per treatment per time-point reached this quality threshold? With only 3 replicates, how many ended up being sequenced for each group? How was statistical analysis considered if the number was too low?

Answer:

In all cases we have ensured to have 3 sequenced replicates for every treatment/time-point, following standard practice in the field, based on a Bayesian criterion supported by B-statistic distributions. By following this guideline in the case a well known organism sequenced by a reliable technology (Illumina NextSeq protocols) statistical robustness enough to allow for differential expression analysis of coding transcripts is expected.

-Line 231: 10-15 million reads doesn’t appear to be deep. What is the rationale for this depth?

Answer:

The number of reads required for RNA-Seq will depend on how sensitive the experiment needs to be, the complexity of your organism and the project goals (in our case, determining differential gene expression of coding genes, without splice variant analysis, in a well known model organism with a mature technology). In general 5 M mapped reads is a good bare minimum for a differential gene expression (DGE) analysis in humans and mammals. In many cases 5 M – 15 M mapped reads are sufficient. This allows us to get a good snapshot of highly expressed genes. A higher sequencing depth generates more informational reads, which increases the statistical power to detect differential expression also among genes with lower expression levels. 

See for instance, Liu, Y., Zhou, J. & White, K. P. RNA-seq differential expression studies: more sequence or more replication? Bioinformatics 30, 301–304 (2014).

- Line 237: how did authors test if the biological processes were indeed active? Mere enrichment of a process among DEGs doesn’t mean active.

Answer:

The reviewer's query regarding the assessment of activity for biological processes is pertinent, considering that mere enrichment among Differentially Expressed Genes (DEGs) doesn't inherently imply activity. While acknowledging the limitations of protein-level measurement within our study, we focused on gene expression as a proxy for activity, recognizing its coarse-grained yet generally reliable approximation. Despite caveats such as the discrepancy between gene expression and protein concentration, and variability in overall proteic activity, high gene expression levels within our study suggest active transcriptional processes driving these genes. Thus, while gene expression serves as an imperfect marker for activity, it remains a valuable tool for assessing the dynamics of biological processes under investigation in our study.

- Lines 320-350: These two sections are confusing and not well justified. Why did authors run these comparisons? What physiological relevance do these comparisons address? While the first section title suggests “contrast between 6 and 12h”, the text seems to describe contrasts among FSH glycoforms at each time-point! The second section title suggests contrasts among each FSH glycoforms, but the text seems to describe how one glycoform is different from others.

Answer:

The reviewer's concerns regarding the clarity and justification of the two sections are valid, and we appreciate the opportunity to provide clarification. What we are presenting here aligns with the concept commonly known as a "difference in differences" comparison. This method entails examining changes in outcomes between two or more groups that have been exposed to different treatments over time. In the context of our study, which involved a perturbation assay, we sought to assess whether different perturbogens (in this case, different FSH glycoforms) led to distinct gene expression patterns across time points. Thus, our objective was to evaluate the similarities and differences in gene set expression patterns among the various glycoforms and time points. While the section titles may not have accurately conveyed this intention, we ensured that the text was revised (lines 170 to 176 of Introduction) to provide a clearer explanation of the rationale behind these comparisons and their physiological relevance within the context of our study.

-Lines 357-358: It is not recommended to separate up and downregulated genes to discover enriched BPs and pathways as some of the genes involved in a particular pathway may be upregulated while other of that pathway may be down regulated (as in lines 514-515).

Answer:

Separating a list of differentially expressed genes into two categories, overexpressed and underexpressed, to perform functional enrichment analysis on each list separately can be a valid and useful strategy in certain contexts. However, as the reviewer correctly pointed out, it's important to note that separating differentially expressed genes into two groups may lead to a loss of important information about interactions between these genes and how they collectively contribute to specific biological processes. However, we decided to use those two lists of genes based on the following considerations:

Overexpression and underexpression of genes MAY have different biological effects and contribute to different aspects of cellular function or observed phenotype. Therefore, analyzing these two categories separately can help better understand the underlying mechanisms. It is also worth noting that, by separating differentially expressed genes into overexpressed and underexpressed categories, it is possible to obtain more detailed information about the biological pathways or cellular functions being regulated in each case. This can facilitate the interpretation of functional enrichment analysis results.

Additionally, separating differentially expressed genes into more homogeneous categories can help reduce biological noise in the analysis, which may improve sensitivity to detect significant associations between genes and biological functions. It is also important to compare and contrast the results obtained from overexpressed and underexpressed gene lists to identify common patterns and differences between them. This can provide a more comprehensive understanding of the underlying biological processes.

Based on the above, we decided to construct two parallel analyses, each one for a differential expression trend. The results were broadly analyzed. The findings of the analysis are also consistent with the experimental results and current knowledge.

- Lines 496-556 : Each of the FSH glycoform appears to have regulated dramatically different set of biological processes. FSH18/21 induced steroid biosynthetic process; FSH24 did not but response to cAMP was both up and down-regulated; eqFSH stimulated processes like ovulation as well as ECM organization that is mostly associated with ovulation; recFSH induced angiogenesis and ECM remodeling processes. Of these, only FSH18/21 appears to have regulated biologically relevant process for follicular granulosa cells. Therefore, it is important for authors to first check which glycoform regulates the usual suspects of granulosa cell genes (Cyp19, Lhcgr, Ccnd2) first before starting to compare different glycoforms.

Answer:

We have checked for Cyp19a1, Lhcgr, and Ccnd2 transcript expression, and the levels of log fold change for the former genes are now included in the supplementary tables. See also the below answer on the expression of these particular transcripts.

- Authors should first prove that all FSH treatment stimulated Cyp19a1 and Lhcgr in granulosa cells, to ensure that results obtained using the cell culture model of this study are relevant to in vivo biology.

Answer:

This reviewer’s concerns are quite pertinent and we apologize for the omission of mentioning these important genes. Fold increase in these genes expression are now included in the supplementary tables. It was interesting to note that mainly after 12 h of exposure to the pituitary glycoforms was fold increase in the Lhcgr and Cyp191a genes clearly detectable. This may be explained by the fact that estradiol production (and presumably Cyp191a mRNA expression) in culture conditions similar to those employed in the present study, occurs late during the incubation period (24-48 h) (Parakh et al., PNAS 103 (33):12435, 2006; Jia and Hsueh, Endocrinology 119(4):1580, 1976). A similar expression profile has also been observed for the Lhcgr (Shi and Segaloff, Mol Endocrinol 9:734, 1995; Gulappa et al., Endocrinology 158(8):2672, 2017). In fact mRNA expression of Lhcgr occurs quite late during the incubation period (peak levels at 48 h; Shi and Segaloff, Mol Endocrinol 9:734, 1995), which may explain the relatively modest fold increase identif

---

## [Decision Letter · Decision Letter 1]

17 Apr 2024

Differential effects of follicle-stimulating hormone glycoforms on the transcriptome profile of cultured rat granulosa cells as disclosed by RNA-seq

PONE-D-23-33249R1

Dear Dr. Alfredo,

We’re pleased to inform you that your manuscript has been judged scientifically suitable for publication and will be formally accepted for publication once it meets all outstanding technical requirements.

Kind regards,

Satish Rojekar, Ph.D.

Academic Editor

PLOS ONE

Reviewers' comments:

Reviewer's Responses to Questions

**Comments to the Author**

1. If the authors have adequately addressed your comments raised in a previous round of review and you feel that this manuscript is now acceptable for publication, you may indicate that here to bypass the “Comments to the Author” section, enter your conflict of interest statement in the “Confidential to Editor” section, and submit your "Accept" recommendation.

Reviewer #1: All comments have been addressed

Reviewer #3: All comments have been addressed

2. Is the manuscript technically sound, and do the data support the conclusions?

Reviewer #1: Yes

Reviewer #3: Yes

3. Has the statistical analysis been performed appropriately and rigorously? 

Reviewer #1: Yes

Reviewer #3: Yes

4. Have the authors made all data underlying the findings in their manuscript fully available?

Reviewer #1: Yes

Reviewer #3: Yes

5. Is the manuscript presented in an intelligible fashion and written in standard English?

Reviewer #1: Yes

Reviewer #3: Yes

6. Review Comments to the Author

Reviewer #1: All my comments have been adequately addressed. The revised manuscript reads well. The conclusions are sound based on the data presented.

Reviewer #3: In This work, Alfredo and colleagues have presented the data to provide information regarding the Importance of glycosylation/ glycoforms of the glycoprotein harmones through Next generation RNA sequencing analysis. The study was well designed and executed. The most important finding of the study include

1. Analysis of the effect of FSH glycoforms in all three possible ways like variations in gene expression, variations in the biological processes and the pathways affected. As these are the 3 most important parameters to delineate any pathways or ligand-receptor interactions of a biological molecules. Authors are succeeded in these by the design of the experiment.

2. This study provides a useful information for further investigation of the therapeutic molecules effecting the signaling pathways in various diseases.

3. Authors are provided the sufficient scientific explanation for the methodology and the results obtained.

7. PLOS authors have the option to publish the peer review history of their article (what does this mean?). If published, this will include your full peer review and any attached files.

Reviewer #1: No

Reviewer #3: No

---

## [Editor Report · Acceptance letter]

20 May 2024

PONE-D-23-33249R1 

PLOS ONE

Dear Dr. Ulloa-Aguirre, 

I'm pleased to inform you that your manuscript has been deemed suitable for publication in PLOS ONE. Congratulations! Your manuscript is now being handed over to our production team.

Kind regards, 

on behalf of

Dr. Satish Rojekar 

Academic Editor

PLOS ONE